# Herbicide Glyphosate: Toxicity and Microbial Degradation

**DOI:** 10.3390/ijerph17207519

**Published:** 2020-10-15

**Authors:** Simranjeet Singh, Vijay Kumar, Jatinder Pal Kaur Gill, Shivika Datta, Satyender Singh, Vaishali Dhaka, Dhriti Kapoor, Abdul Basit Wani, Daljeet Singh Dhanjal, Manoj Kumar, S. L. Harikumar, Joginder Singh

**Affiliations:** 1Department of Biotechnology, Lovely Professional University, Phagwara 144411, India; simnav14@gmail.com (S.S.); dhakavaishali@gmail.com (V.D.); daljeetdhanjal92@gmail.com (D.S.D.); 2Punjab Biotechnology Incubator (PBTI), Phase-V, S.A.S. Nagar, Punjab 160059, India; 3Regional Advance Water Testing Laboratory, Department of Water Supply and Sanitation, Phase-II, S.A.S. Nagar 160054, India; satyenderjamwal@gmail.com; 4Regional Ayurveda Research Institute for Drug Development, Gwalior 474009, India; vkumar8491@gmail.com; 5Department of Education, Adelaide, SA 5000, Australia; jpkaur015@gmail.com; 6Department of Zoology, Doaba College Jalandhar, Jalandhar 144001, India; shivikadatta@gmail.com; 7Department of Botany, Lovely Professional University, Phagwara 144411, India; dhriti.21851@lpu.co.in; 8Department of Chemistry, Lovely Professional University, Phagwara 144411, India; basitwani81@gmail.com; 9Department of Life Sciences, Central University Jharkhand, Brambe, Ranchi 835205, India; manoj@cuj.ac.in (M.K.); slharikumar@gmail.com (S.L.H.)

**Keywords:** herbicide, glyphosate, toxicity, environmental fate, biodegradation, catabolic enzymes

## Abstract

Glyphosate is a non-specific organophosphate pesticide, which finds widespread application in shielding crops against the weeds. Its high solubility in hydrophilic solvents, especially water and high mobility allows the rapid leaching of the glyphosate into the soil leading to contamination of groundwater and accumulation into the plant tissues, therefore intricating the elimination of the herbicides. Despite the widespread application, only a few percentages of the total applied glyphosate serve the actual purpose, dispensing the rest in the environment, thus resulting in reduced crop yields, low quality agricultural products, deteriorating soil fertility, contributing to water pollution, and consequently threatening human and animal life. This review gives an insight into the toxicological effects of the herbicide glyphosate and current approaches to track and identify trace amounts of this agrochemical along with its biodegradability and possible remediating strategies. Efforts have also been made to summarize the biodegradation mechanisms and catabolic enzymes involved in glyphosate metabolism.

## 1. Introduction

Glyphosate [N-(phosphonomethyl) glycine CAS#1071-83-6] is one of the most extensively used broad-spectrum organophosphorus herbicides [1]. It is a widely used herbicide in agriculture against perennial and annual weeds and in silviculture, domestic gardens, and urban areas [2]. It is an essential component of non-selective and post-emergent herbicides used to protect the crop from grasses, annual broad-leaved weeds, woody plants, etc. [3]. The parent compound was firstly sold in 1974 under the trade name “Roundup” by Monsanto [4]. This compound tends to be a zwitterion, in which phosphonic hydrogen detaches and joins the amine group. Glyphosate was first synthesized by Henri Martin while working at Cilag (a Swiss pharmaceutical company), but J.E. Franzo in 1970 conducted the herbicidal test on this compound and commercialized it in 1974 [5]. The potential mode of action of glyphosate makes it an herbicide of interest. The global glyphosate market was $23.97 billion in 2016, and at a growth rate of 6.05% for the forecasting period, it is estimated to reach $34.10 billion in 2022 [6]. 

Glyphosate is the only herbicide that targets 5-enolpyruvylshikimate-3-phosphate synthase (EPSPS) without any available analog and obstructs the aromatic amino acid biosynthesis in the shikimate pathway [7]. Inhibition of EPSPS by glyphosate retards the synthesis of essential secondary metabolites and proteins; additionally, it curbs the vital energy pathways in soil microbes and plants [8]. A study reveals that glyphosate alters the soil texture and microbial diversity by reducing the microbial richness and increasing the population of phytopathogenic fungi [9]. This herbicide is considered safer than others, but its overuse imposes chronic effects on the environment and humans [10]. Moreover, its broad herbicidal activities and the development of transgenic glyphosate-resistant crops (e.g., cotton, canola, maize, and soybean) are the main reasons for the excess use of this herbicide [11]. Unawareness regarding the use of this herbicide has led to its accumulation in both terrestrial and aquatic ecosystems [12]. As glyphosate can be absorbed by soil particles, it often remains at the vadose zone [4]. Hence, it is usually detected in surface water, water-sediment interface from surface run-off, and groundwater [13]. 

The International Agency for Research on Cancer (IARC) classified glyphosate as “Category 2a,” which specifies probable carcinogenic to humans [14]. The United States Environmental Protection Agency (USEPA) classifies this herbicide as “Group E carcinogen,” which means non-cancerous for humans [4]. In contrast to the European Food Safety Authority, which determines glyphosate as a potent carcinogen for humans, but the experimental evidence does not support this determination [4], even though traces of glyphosate have been detected in human urine samples, highlighting its persistence, bioaccumulation, and potential health risk [15]. Although glyphosate residual concentrations have never crossed over the threshold level, its harmful effects cannot be ignored [16]. A study reveals that this herbicide alters the soil texture and microbial diversity by reducing the microbial richness and increasing the population of phytopathogenic fungi [17]. Literature has proved that glyphosate is among the carcinogenic compounds and can cause organ failure by inhibiting acetylcholinesterase and inducing oxidative stress in non-mammalian species [9]. Aminomethylphosphonic acid (AMPA) is the keystone metabolite of glyphosate often found in the sediment, surface, and groundwater [4]. Various in vitro toxicity studies have disclosed that AMPA affects human red blood cells and can lead to chromosomal aberrations in fish [18]. 

Degradation of glyphosate can be achieved using abiotic and biotic means, e.g., absorption, photolysis, thermolysis, and biodegradation with catabolic enzymes. Lately, a blend of photocatalyst with UV light has come in the limelight for their ability to treat pollutants like pesticides. The photocatalytic degradation can break glyphosate down to non-toxic compounds like CO_2_, water, and inorganic ions [4]. This mechanism depends on the photocatalytic oxidation reaction triggered by highly reactive oxidation and hydroxyl radical [19]. The benefits of photocatalysis comprise cost-effectiveness, stability, and non-toxicity, whereas its manipulation in situ is a critical disadvantage. This degradation method proved to be optimal for removing glyphosate in sewage treatment [10]. An eco-friendly strategy like bioremediation would be another promising alternative to overcome the environmental and health risks derived from glyphosate and its residues. Therefore, it has become essential to study glyphosate biodegradation driven by microbial degraders. Numerous studies have revealed microbial capacity as a robust and useful tool in bioremediation. However, previously published literature fails to comprehend the mechanisms and pathways by which different microbial species can degrade glyphosate. This review aims to outline the ecotoxicity of glyphosate and its biodegradation via the roles of catabolic enzymes.

## 2. Glyphosate Toxicity

### 2.1. Ecotoxicity

The intensive use of glyphosate has led to the contamination of different ecosystems adversely affecting animals, plants, and microorganisms, which further leads to the deterioration of food chains. The pesticide residues often remain in the food chain, and diet contains nutritional deficiencies, especially vitamins and minerals and has the capability to incur systemic toxicity. Some studies revealed that glyphosate is becoming a major issue in the growth of chronic diseases. Glyphosate strongly disrupts soil biology as it is toxic to beneficial microflora and earthworms. Being a potent inhibitor of EPSPS in the shikimic acid pathway, it exerts adverse effects on non-target aquatic plants, which are a great environmental concern (Figure 1). The functional characterization of the plants is also affected by glyphosate as PGPRs play a vital role in the production of auxin and siderophores, solubilization of phosphate, and zinc uptake which convert insoluble forms of macronutrient elements into soluble forms. 

The United States environmental protection agency (USEPA) considers glyphosate in toxicity class–IV for inhalation and also reports its relatively low oral and dermal acute toxicity [4]. It can cause irritation, vomiting, nausea, and photo contact dermatitis [20]. It is slightly toxic for amphibians and fishes. It is excreted in urine and feces [4]. It is less persistent in water and has a half-life of 12 days to 70 days [21]. It is reported to bio-accumulate in animals and has the ability to break down considerably depending on the particular environment [4].

Glyphosate alters several plant physiological aspects due to its herbicidal appearance. A number of off-target plants species have been studied so far to evaluate the adverse effects of glyphosate, viz., *Zea mays, Oryza sativa, Tritium aestivum,* and *Pisum sativum* [22,23,24,25,26]. Various physiological processes in the plants show the negative impacts of glyphosate toxicity including oxidative burst, drop in the synthesis of chlorophyll [26,27,28], hampered photosynthetic rate, alteration in the level of plant hormones, reduced nitrogen assimilation, decreased nutritional content, disturbances in lignin and carbon metabolism [23]. It affects photosynthetic activity indirectly by obstructing the synthesis of pigments and fatty acids, decreasing the stomatal conductance, and declined the availability of PSII-associated metals and amino acids, which ultimately reduces its ability to transfer photo-chemical energy into the electron transport chain [24]. Oxidative stress caused by glyphosate is due to the overaccumulation of reactive oxygen species (ROS) and disturbance in the mineral nutrients level. It inhibits the EPSPS enzyme, which ultimately blocks the shikimate pathway and results in inhibiting the biosynthesis of plant secondary metabolites [29]. It also alters the nitrogen metabolism by affecting the rhizobial symbiont or indirectly by altering the physiology of the host genotypes [30]. As a metal chelator, glyphosate declines level of vital nutrients having key roles as enzymatic co-factors and biomolecular constituents (Figure 2). 

Production of quinone (a secondary metabolite and an integral part of various biochemical processes), chlorophyll, fatty acids, and amino acids is inhibited as an aftermath of its overuse [30,31,32]. Reduction in chlorophyll level and hence the photosynthetic efficiency are attributed to inhibition in the amount of Mg in leaves [30] along with that of Fe leading to the prevention of biosynthesis of significant chlorophyll enzymes biosynthetic pathway like catalase, peroxidase and δ-aminolevulinic acids [33]. Working of enzymes- ascorbate peroxidase (APX), catalase (CAT), and polyamine (PA) in *Lemna minor* tissues is also hampered by glyphosate-containing herbicides [34]. Glyphosate also competes with amino acid glycine resulting in loss of glutamate or with a principal product of alanine synthetase active site hindering the synthesis of alanine in C_2_ cycle/photorespiration [35]. 

An assessment of glyphosate’s spray retention, uptake by leaves and its translocation in *Ambrosia artemisiifolia* revealed that it translocates to the growing apical tissues through the roots quite rapidly almost within three hours [36]. Accessibility of amino acids, as well as metal ions related to the PSI and PSII to the transmission of light energy into the cyclic and non-cyclic electron transport chain systems, is notably depleted by glyphosate [37]. CO_2_ assimilation potential of plants by a reduction in net carbon exchange and stomatal conductance get adversely affected by foliar spray of glyphosate and its metabolites [29,38]. Improper functioning of enzyme Rubisco owing to the dropping down of levels of ribulose-1,5-biphosphate (RuBP) and 3-phosphoglyceric acid (PGA) in the Calvin cycle is also linked to glyphosate toxicity [39].

Physiological efficiency of the host plant is dwindled by glyphosate via indirectly disturbing the nitrogen metabolism or directly harming the symbiotic rhizobial bacteria present in the soil [29] due to which stoppage of growth and ultimately death may occur [39]. Further studies showed a reduction in nodule formation and nitrogen fixation process by the incessant use of this very herbicide [30]. Induction of nutritional distress in plants is another harmful upshot of glyphosate, arising because of the latter’s interference with the location mechanism of nutrients. Reports suggest that obstructions are appearing in the shikimate pathway cause oxidative stress by inhibition of particular target sites of the plants in which oxidative stress markers are seen to undergo changes in their nature [23]. Morphological and operational expertise exhibited by plants is also put at stake by glyphosate because of decline in lignin content [40]. Suppression of EPSPS and reduced availability of cinnamate precursors contribute significantly towards lowering down of the extent of lignin synthesis [41]. Hormonal balance in *Glycine max* gets distressed due to glyphosate, which in turn mar its growth and development features [42]. Blockage of the shikimate pathway, which arrests the auxin synthesis from its indolic precursor tryptophan, is likewise found to be associated with glyphosate. Harmful impacts of glyphosate use on algae have also been affirmed through estimation of measured environment concentrations (MEC) and the EC_50_ value of glyphosate in *Scenedesmus quadricauda,* which turned out to be 0.1 µg/L and 4.4 mg/L, respectively [43].

The noxious effects of herbicides are not only limited to unicellular organisms but also in multicellular organisms [4]. Right from the lower invertebrates up to higher chordates, all the animals are found to be affected by this herbicide. Invertebrates like earthworms are majorly affected as determined by some parameters like a reduced rate of reproduction, loss of biomass, DNA damage and reduced casting activity [44,45,46]. Glyphosate and its formulation Roundup were used for toxicity evaluation on two different populations of *Daphnia magna* (one population was taken from laboratory cultures and other was wild-collected species). Roundup showed acute toxicity values after 48 h in the range 560-1700 μg/mL. The wild species was found to be twice more sensitive towards Roundup than glyphosate. However, the standard species depicted Roundup being 35 times more toxic than glyphosate [47]. Gaur and Bhargava [48] reported acute toxicity to zebrafish embryos exposed to glyphosate concentration at 50 µg/mL and above. Developmental toxicity was characterized by physical malformations in zebrafish embryos such as yolk sac edema, pericardial edema, eye defects body bending, etc. The LD_50_ for glyphosate was reported 66.04 ± 4.6 µg/mL after 48 h of exposure [48]. There is also a report that glyphosate adversely affects the spermatozoa of the honey bee. After exposure of glyphosate for 40 min, LD_50_ concentration was found to be 0.31 mg/mL (*p* < 0.0001). The lethal time (LT_50_) at 0.05 mg/mL concentration of glyphosate was found to be 468 min (*p* = 0.009) [49]. The effect of herbicide, glyphosate on Wistar rats was evaluated in a 28 d study. Daily dose equivalent to 0.1 acceptable operator exposure level (AOEL) and 0.5 consumer acceptable daily intake (ADI) was considered for evaluation. The treated animals weighed less in comparison to control. Treatment groups had significant primary DNA damage in the liver cells and leukocytes. Acetylcholinesterase (AChE) activity was found to be inhibited with all the treatments suggesting that even low doses of glyphosate exposure can cause serious adverse effects [50]. Glyphosate undergoes one-compartment model with an elimination rate constant of K_el_ = 0.021 h^−1^ when 100 mg/kg bw) of glyphosate concentrations in plasma in Sprague-Dawley rats [51].

The leaching of glyphosate in aquatic systems reduced the egg-laying capacity and hampered the hatching process in many aquatic animals like sea urchins and snails [52]. Apart from lower organisms, glyphosate is also known to exert toxic effects on humans. It is a potential endocrine disruptor in human beings which causes serious damage to the placenta [52]. It is also genotoxic and causes plasma damage and epithelial cell damage in humans. Hepatic, embryonic and placental cell lines are known to be affected largely by this herbicide. It affects almost the entire animal kingdom. It affects the food chain and causes undesired changes that impose serious concern.

The presence of glyphosate in food, water, and air confirms that it is ingested frequently and found in human urine at levels around 110 µg/L [13]. AMPA and glyphosate itself are the major metabolites detected in human urine samples [15,53,54]. The maximum contamination level (MCL) of glyphosate is reported to be 700 μg/L in the USA (EPA 2015) and 1000 μg/L in Australia [55]. In Europe, the tolerable risk is reported to be 77 μg/L [56] while the acceptable concentration in drinking water is less than 0.1 μg/L.

### 2.2. Cytotoxicity and Genotoxicity

The study of Roundup on human cell cultures depicts an increase in Cytochrome P 450 (CYP 450) activity [57]. The formulation (Roundup) was reported to cause problems during pregnancy as confirmed by exposing the JEG3 (human placental cell line). The study also reported the inhibition of aromatase enzymes and fluctuated mRNA [58]. The effect of Roundup Ultra 360 SL, along with glyphosate on human erythrocytes, depicted the elevation in the level of methaemoglobin and hemolysis but showed no significant change in GSH (glutathione level) [59]. Glyphosate exposure is also reported to cause arrhythmia, hypotension, mental relapse, and renal and respiratory failure in humans [60]. A cytotoxic effect like membrane damage as well as impaired mitochondrial function, which may lead to cancer on the treatment of dosage less than 40 mg/L to buccal epithelial cell line TR146. However, on the dose of >80 mg/L, elevated lactate dehydrogenase (LDH), leading to membrane and DNA damage, was observed [61]. The intoxication of glyphosate is reported to cause cardiovascular shock, hemodynamic hindrance, intravascular coagulation, myocardial infarction and multiple organ failure [62]. In vivo glyphosate toxicity assessment over HepG2 (hepatic cell line) revealed the disruption of an androgen receptor (MDA-MB453-kb2) and termination of the transcription cycle of estrogen receptor of HepG2 at 0.5 ppm. At 5 ppm, further DNA damage was observed, and at 10 ppm concentration, it included cytotoxic effects as well [57]. Glyphosate based herbicides (GlyBH) have been reported to cause chronic effects such as hepatorenal, teratogenic, and tumorigenic effects along with endocrine functionality disruption [14]. Amendment in estrogen response element, which further alters the ERα and ERβ expression up to 513-fold was reported by Thongprakaisang and coworkers [63]. The effect of glyphosate (0.36 mg/L) was investigated on sperm motility and sperm DNA fragmentation. It was found out that 1 h post-treatment, the sperm motility was significantly reduced in comparison to respective control although sperm DNA fragmentation was not significantly different in the first hour [64]. A decrease in peripheral blood mononuclear cells (PBMCs) viability by 2.7% was observed after incubation with glyphosate (19 mM) for 24 h [65]. Glyphosate exposure decreases cell viability depending on concentration and time exposure in human epithelial type 2 cell line (HEP -2) [66]. Glyphosate levels of 1 and 10 exposed to the blood–brain barrier (BBB) in vitro based on induced pluripotent stem cells (iPSCs) depicted an increase in the permeability across BBB [67]. Glyphosate and AMPA concentrations up to 50 mM were observed to inhibit cell growth in eight cancer cell lines, including four prostate cancer (C4-2B, LNCaP, DU-145, and PC-3), two ovarian cancer (SKOV-3 and OVCAR-3), one cervical HeLa, and one lung cancer cell line (A549) [68]. Hoppin et al. [69] reported the association of glyphosate with atopic asthma. However, Henneberger and associates [70] found an inverse association between glyphosate and asthma symptoms depicting that patients with asthma may be less sensitive to glyphosate. 

## 3. Microbial Degradation of Glyphosate

Extensive and unrestricted applications of glyphosate threatened the health of humans, animals, non-target plants, and microbial flora in various ecosystems. Several studies have scrutinized the effect of glyphosate on carbon mineralization. There are three major intermediates of glyphosate metabolism, AMPA, acetyl-glyphosate, and sarcosine, which get further degraded via different metabolic pathways. The primary metabolite of glyphosate is AMPA. It causes secondary contamination in the environment because of the non-feasibility of intracellular degradation. Some studies have shown that glyphosate can ominously increase microbial respiration for the short term, which typically happens from 7 to 38 days. Other studies have specified that glyphosate has no effect on microbial respiration.

Various studies have been reported on the degradation of glyphosate and its metabolites in different environmental matrices (*Achromobacter* sp. MPK 7A, *Comamonas odontotermitis* P2, *Ochrobactrum intermedium* Sq20, and *Pseudomonas* sp. 4ASW) in contaminated niches via enrichment approach [71,72,73,74]. Microbial species such as *Arthrobacter atrocyaneus* ATCC 13752, *Alcaligenes* sp. GL, *Arthrobacter* sp. GLP-1, *Geobacillus caldoxylosilyticus* T20, and *Pseudomonas* sp. PG2982 were reported to consume glyphosate as a nutrient for their growth [75,76,77,78,79]. From the list of glyphosate-degrading microbes, including actinomycetes, bacteria, fungi, and microcytes, the bacteria hold the primary position [16,80,81]. Thus, to assess the glyphosate-degrading ability of microbes for their application in bioremediation, the optimization of the growth conditions like pH, temperature, incubation time, inoculum, and glyphosate concentration is necessary. The results obtained from response surface methodology (RSM) revealed that bacteria are effective in degrading glyphosate when grown under optimum conditions [73,82]. *Arthrobacter* sp. GLP-1, *Alcaligenes* sp. GL, *Pseudomonas pseudomallei* 22, and *Flavobacterium* sp.GD1 utilize glyphosate as a phosphorus source [75,79,83,84].

In the same way, *Ochrobactrum intermedium* strain Sq20, *Agrobacterium radiobacter* strain SW9, and *Achromobacter* sp. strain LW9 use glyphosate as nitrogen or carbon source [74,85]. Various fungal populations like *Aspergillus oryzae* A-F02, *Aspergillus niger*, *Penicillium* IIR, *Mucor* IIIR, *Trichoderma harzianum*, *Scopulariopsis* sp., and *Penicillium notatum* utilize glyphosate as sole phosphorus source [80,86,87,88,89]. Moreover, several strains of microbes utilize glyphosate as a different type of energy source like *Arthrobacter* sp. GLP-1/Nit-1 [90], *Comamonas odontotermitis* P2, and *Streptomycete* sp. StC, which decompose glyphosate for sole nitrogen, sole phosphorus, and both nitrogen and phosphorus source, respectively [73,81]. Another *Arthrobacter* species, *Arthrobacter atrocyaneus* ATCC 13,752, has been reported to rapidly degrade AMPA and glyphosate without enrichment technique [78]. Another bacterial species, *Flavobacterium* sp. GD1, uses AMPA and glyphosate solely as a phosphorus source. Moreover, it has been found that Pi (inorganic phosphorus) concentration does not affect the metabolism of glyphosate, but it suppresses the process of AMPA degradation [79]. Inhibition via Pi has been observed in various bacterial isolates such as *Pseudomonas* sp. GLC11 and *Pseudomonas* sp. PG2982, using phosphorus moiety of glyphosate [76,91,92,93].

To date, there are only three chief intermediates of glyphosate metabolism such as AMPA. Acetyl-glyphosate and sarcosine have been found to get further degraded via different metabolic pathways. AMPA is the primary and major metabolite of glyphosate degradation pathway. Due to the non-feasibility of intracellular degradation of AMPA, it becomes the source of secondary contamination in the environment [75,83,94]. The strains that have been known to utilize AMPA as sole Pi source include *Pseudomonas* sp. 4ASW, *Pseudomonas* sp. LBr, *Bacillus megaterium* 2BLW, and *Pseudomonas* sp. [94,95]. 7B. In contrast to AMPA pathway, the bacterial isolates also degrade glyphosate to sarcosine and use sarcosine as a growth nutrient. Few reports enlighten on the bacterial isolates like *Bacillus* cereus CB4, *Ochrobactrum anthropi* GPK 3, and *Pseudomonas* sp. LBr, which robustly degrade glyphosate to AMPA as well as sarcosine [82,94,96]. Thus, bacterial strains showing the potential of degrading glyphosate emerged as the bioremediation tool for remediating the contaminated environment. On the sequence analysis of C-P lyase (carbon-phosphorus lyase) and glyphosate oxidoreductase genes, it was discovered that both pathways operate in parallel in *Comamonas odontotermitis* P2 [97]. In one of the metabolic processes, glyphosate gets converted into acetylglyphosate but is not further degraded by the *Achromobacter* sp. Kg 16 to use it as a phosphorus source. Unpredictably, the same bacterial isolate degrades glyphosate to AMPA when a carbon source is absent in the culture medium [72].

## 4. Degradation Kinetics and Glyphosate Residues 

Glyphosate is a common herbicide used worldwide, having the capability of rapid degradation in soils, particularly by microbial processes. AMPA is the most commonly produced degraded product of glyphosate in water and soil. Tazdaït and coworkers [98] determined glyphosate residues by colorimetric method and at 570 nm absorbance. They also determined the percentage of glyphosate removal by using the following formula:Percent glyphosate removal = C_i_ − C_f_/C_f_ × 100(1)
where C_i_ and C_f_ are the glyphosate concentrations values at times 0 and t, respectively. 

Various kinetic models and adsorption isotherms were found best fit to the biodegradation kinetics of glyphosate like Andrews [99], Aiba et al. [100], Han and Levenspiel [101], Luong [102], Tessier [99], Webb [99], Tseng and Wayman [103], and Yano and Koga [104]. However, from the biodegradation point of view, pseudo order kinetics was the best fit model for degradation studies. Biodegradation rate can be determined by using pseudo-first-order kinetics as reported in our previous studies showing 85-90% of glyphosate with half-life periods from 8.36 to 9.12 days with major metabolites included glycine, AMPA, sarcosine, glyoxylate, metaphosphoric acid, and phosphate [105]. The equation below correspond to the equations of the concentration variation of glyphosate with time plotted (t) on the basis of the linear regression results obtained by plotting Time (in days) vs. Log Ct (in ppm).
d[C]/dt = −k_obs_[C](2)
log[C]/[C]_0_ = k_obs_t(3)
t_1/2_= (1/k_obs_) × log 2(4)
where, [C] = glyphosate concentration at time t (ppm); [C]_0_ = the initial glyphosate concentration (ppm); k_obs_ = the pseudo-first-order constant (day^−1^) which is equal to slope of line, i.e., k_obs_ = −slope.

## 5. Mechanism Underlying Bio-Degradation

Glyphosate can be easily degraded by two metabolic pathways, i.e., AMPA and sarcosine via glyphosate degrading bacteria. This is achieved by (a) oxidase, which degrades the carboxymethylene-nitrogen bond of glyphosate and converts it into AMPA and glyoxylate and/or (b) C-P lyase, which directly cleaves the carbon-phosphorus bond to produce sarcosine. Both degradation pathways use C-P lyase to cleave the C-P bond of AMPA compound.

The primary step involved in glyphosate degradation pathway is the catalytic action of glyphosate oxidoreductase, which synthesizes the glyoxylate and AMPA. *Pseudomonas* sp. LBr degrades the glyphosate through AMPA and glycine pathway, and on NMR analysis, it was found that it uses formaldehyde and glyoxylate for its development [94]. Another bacterial strain, *Arthrobacter atrocyaneus* ATCC 13,752 degrades glyphosate into AMPA and then finally to CO_2_, but CO_2_ obtained was not the by-product of AMPA [90]. The AMPA, an intermediate metabolite, gets liberated into the ecosystem or further degrades via different enzymes [78,94,96]. Generally, AMPA act as a substrate for enzyme C-P lyase, converting it to P_i_ and methylamine for phosphorus uptake for *Arthrobacter* sp. GLP-1, *Arthrobacter atrocyaneus* ATCC 13752, and *Pseudomonas* sp. LBr [71,78,106]. Lately, *Ochrobactrum anthropi* GPK3 isolate illustrated the new AMPA degradation pathway in which AMPA is first degraded to phosphonoformaldehyde via transaminase enzyme and then further to formaldehyde via phosphonatase (Figure 3). 

Another glyphosate degradation pathway involves C-P lyase and leads to the synthesis of P_i_ and sarcosine, for example, *Pseudomonas* sp. PG2982 metabolizes glyphosate via C-P lyase and produces sarcosine, which further gets degraded to formaldehyde and glycine by the action of sarcosine oxidase [91]. It has been found that *Arthrobacter* sp. GLP-1 uses glycine for the protein synthesis by stimulating the formation of amino acids (serine and threonine) and peptide backbone [78]. This fate of glyphosate metabolites has been noticeably assessed with the help of isotope labelling. 

## 6. Enzymatic Mediated Metabolism of Glyphosate

Glyphosate oxidoreductase (GOX) is responsible for the degradation of glyphosate to AMPA by cleaving the C-N bond in bacterial species [4]. On screening the sequence of *Comamonas odontotermitis* P2(KX980206.1), *Ochrobactrum anthropi* GPK3, and *Ochrobactrum* sp. G1 (GU214711.1), it has a similarity index of 99% similarity with the GOX-encoding genes. Hence, the synthetic construct between GOX gene (HQ110097.1) and GOX (ADV58259.1) has been developed so that it can be used for developing a transgenic glyphosate-tolerant variety of canola [97,107]. However, GOX enzyme (purified) obtained from microbes or synthesized by expressing the *gox* gene in expression vector shows the low affinity towards glyphosate [108,109]. *Ochrobactrum anthropi* GPK 3 producing GOX enzyme containing flavin adenine dinucleotide (FAD) confirms its relation with flavin monooxygenase superfamily [15]. glpA and glpB, are the two ORFs (Open reading frames) in *Pseudomonas pseudomallei* 22, which are related to glyphosate metabolism. glpA provides tolerance against glyphosate, and glpB is linked with the degradation of glyphosate to AMPA [84].

Enzymes like C-P lyase, phosphoenolpyruvate hydrolase phosphonoacetatehydrolase, and phosphonoacetaldehyde hydrolase have been reported in the lysis of C-P bond of glyphosate [110], where C-P lyase disintegrates the glyphosate C-P bond while remaining are site-specific with their substrate, C-P bond of glyphosate being hydrolytically stable and highly resistant to photolysis and chemolysis. Only C-P lyase has a high affinity for glyphosate which breaks the C-P bond (inactive form) resulting in the formation of sarcosine. Therefore, C-P lyase complex has been extensively studied in *E. coli.* In *E. coli*, this complex is formed by the product of 14 operon genes (*phn* CDEFGHIJKLMNOP), which are related to Pho regulon. In accordance with previous biochemical and genetical studies, it has been confirmed that *phn*CDE encodes for transporter protein of ATP-binding cassette, and *phn*F encodes for repressor protein [111,112,113], while seven proteins (*phn*G-I; K-M) are considered to be encoding for core components of C-P lyase (membrane-bound), which are involved in degrading phosphonates into phosphate, except *phn* which acts as the catalyst [113,114].

Furthermore, *phn*NOP performs the accessory and regulatory function in catabolic degradation pathway involving C-P lyase, whereas pure and well-characterized C-P lyase cannot degrade the C-P bond of glyphosate. Until now, C-P lyase, having a high affinity towards glyphosate, has not been characterized at the molecular level. Moreover, we have found another C-P lyase which is non-specific to glyphosate but is involved in the C-P bond cleavages during the degradation of AMPA. This evidence proves that one bacterium probably has two co-existing C-P lyases having varied substrate specificity [113,114]. The detailed enzyme-mediated biodegradation of glyphosate has been outlined in Figure 4.

Usually, glyphosate metabolism for AMPA synthesis is not dependent on P_i_ concentration, excluding *Arthrobacter atrocyaneus* ATCC 13752, as P_i_ represses the degradation of glyphosate [78]. P_i_ concentration regulates the metabolic conversion of glyphosate to sarcosine, e.g., repression of glyphosate degradation was observed in *Arthrobacter* sp. and *Pseudomonas* sp. 4ASW in the presence of P_i_ [71,106]. Their glyphosate transporter system revealed that they are dependent on P_i_ level. Additionally, biochemical and molecular studies revealed that C-P lyase is regulated by *phn* genes, which gets up-regulated during the absence of P_i_. Along with that, the role of the two-component system (PhoR-PhoB) and their responses to the endogenous and exogenous P_i_ concentration have been studied in *E. coli*. Hence, it can be stated that the P_i_ level hinders the sarcosine pathway of glyphosate metabolism [114,115]. Table 1 outlines various microbial species responsible for glyphosate metabolism and enzymes responsible for degradation processes.

## 7. Conclusions

The review portrays that there is much more to be learned about the fate of glyphosate, including its sorption, degradation, and leaching. The fate depends upon the medium and varies a lot from soil to soil as well. This variability does not give a clear prediction, and results generate ambiguous conclusions. Apart from environmental risks, glyphosate is also associated with health risks. This makes for the requirement to develop an eco-friendly strategy for bioremediation. Glyphosate being a potent inhibitor of EPSPS in the shikimic pathway, exerts negative effects on non-target plants. Various ecosystems and their abiotic and biotic components including animals, plants, and microbes are adversely affected by the indiscriminate use of glyphosate. Right from unicellular to multicellular and lower to higher invertebrates, glyphosate affects all the animals from the kingdom. Right from the lower invertebrates up to higher chordates, all the animals are found to be affected by this herbicide. Glyphosate can actually be degraded via two metabolic pathways—AMPA and sarcosine through glyphosate degrading bacteria. Soil organic matter also indirectly affect the sorption of glyphosate. The phosphate in soil competes with glyphosate for sorption, which ultimately affects the retention and degradation of glyphosate. The pre-sorption of phosphates almost eliminates the chances of glyphosate sorption in some soils. Though in some soils the availability of phosphate is found to accelerate the degradation of glyphosate. There is still more to it, and this gap needs to be studied further. The fate of glyphosate should be considered for studies as it is primary for environmental and health risk assessments. For optimal use of glyphosate, social costs, in addition to direct costs, should also be taken under consideration. Glyphosate is found to be an absolutely effective herbicide and should be considered for the development of new cost-effective herbicides for sustainable agriculture. There is no question that glyphosate resistant crops make glyphosate the most inexpensive and effective technology for weed management. But after its advent, with time, its effectiveness decreased and the mineral nutrition was also altered due to the capability of glyphosate to chelate with metal ions. Certain additives may be used to enhance the biological performance of glyphosate. A better understanding of glyphosate action and usage of adjuvants for overall better effect should be other approaches to designing future glyphosate formulations. There is still more to it, and this gap needs to be studied further.

## Figures and Tables

**Figure 1 ijerph-17-07519-f001:**
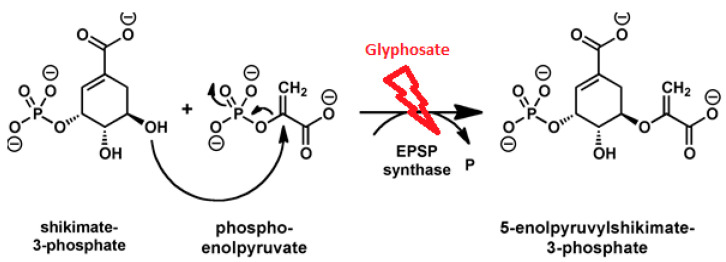
Schematic representation of binding of glyphosate to the substrate-binding site of the EnolPyruvylShikimate-3-Phosphate Synthase (EPSPS), inhibiting its activity, and blocking its importation into the chloroplast.

**Figure 2 ijerph-17-07519-f002:**
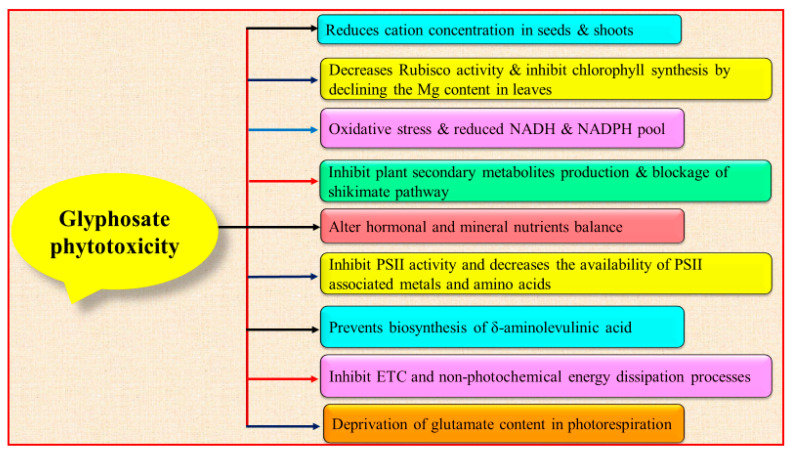
Glyphosate toxicity mediated alteration in the various physicochemical functions of plants (NADH = Nicotinamide adenine dinucleotide hydrogen NADPH = Nicotinamide adenine dinucleotide phosphate, PSII = Photosystem II, ETC = Electron Transport Chain).

**Figure 3 ijerph-17-07519-f003:**
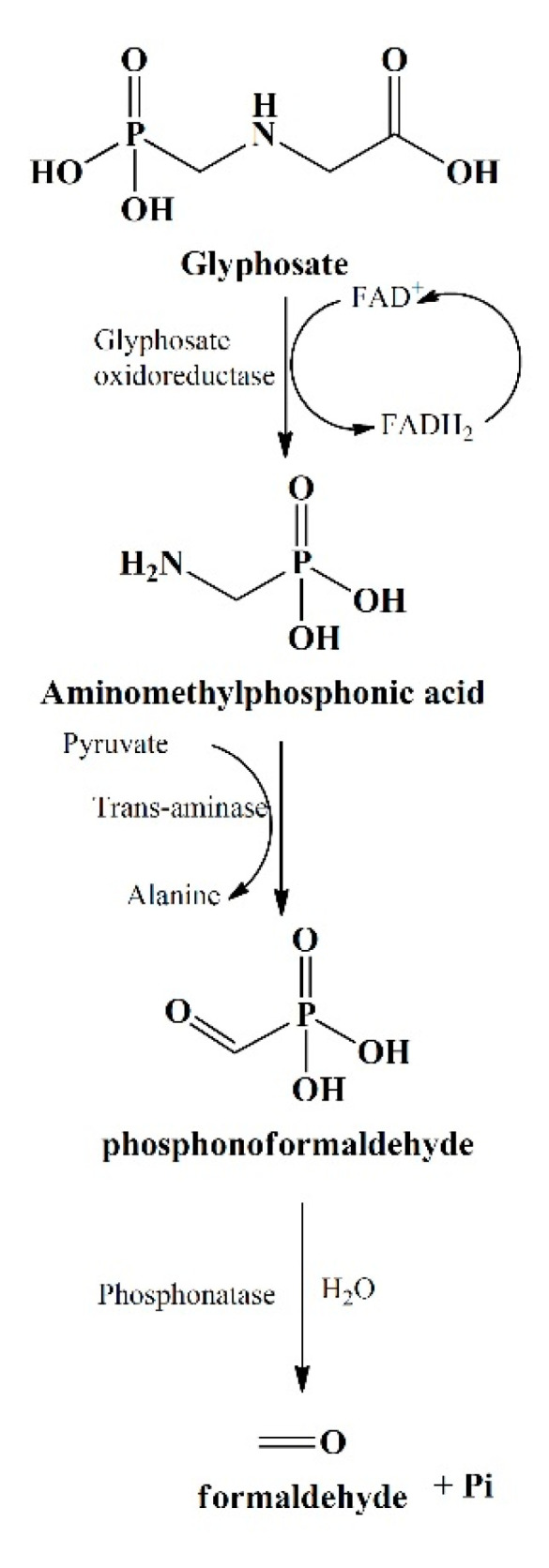
Glyphosate bio-degradation pathway via Aminomethylphosphonic acid (AMPA) and formaldehyde formation.

**Figure 4 ijerph-17-07519-f004:**
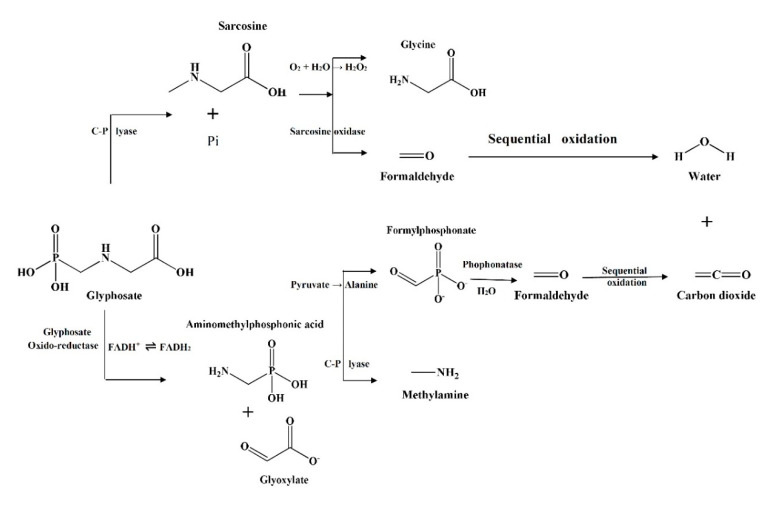
Enzyme mediated biodegradation of glyphosate.

**Table 1 ijerph-17-07519-t001:** A list of various microbial species responsible for glyphosate metabolism and enzymes responsible for degradation processes.

Microbial Species	Source	Enzymes/Gene	Location	Metabolites after Degradation	References
*Ochrobactrum anthropi* GDOS	Soil	-	Iran	AMPA	[16]
*Pseudomonas* sp. 4ASW	Soil	-	UK	Sarcosine	[71]
*Comamonas odontotermitis* P2	Soil	Glyphosate oxidoreductase	Pakistan	-	[73]
*Alcaligenes* sp. GL	Selective medium	-	Germany	AMPA (5%) and sarcosine (95%)	[75]
*Pseudomonas* sp. PG298231	Mixed culture	C-P lyase, sarcosine oxidase	Louisiana	Sarcosine	[76]
*Geobacillus caldoxylosilyticus* T20, *Ochrobactrum anthropi* LBAA	Soil		UK	AMPA	[77]
*Arthrobacter atrocyaneus* ATCC 13752	Microbial collection	C-P lyase	Germany	AMPA	[78]
*Arthrobacter* sp. GLP-1	Selective medium	-	USA	Sarcosine	[79]
*Penicillium notatum*	Mutation of the wild type	-	Poland	AMPA	[80]
*Streptomyces* sp. StC	Sludge	-	Poland	Sarcosine	[81]
*Bacillus cereus* CB4	Soil	-	China	AMPA, glyoxylate, sarcosine, glycine and formaldehyde	[82]
*Flavobacterium* sp. GD1	Sludge	-	Missouri	AMPA	[83]
*Pseudomonas pseudomallei*	Soil	glpA and glpB	USA	AMPA	[84]
*Achromobacter* sp., *Rhizobium radiobacter*	Sludge	-	USA	AMPA	[85]
*Aspergillus oryzae A*-F02	Soil	-	China	AMPA and methylamine	[86]
*Aspergillus niger*, *Scopulariopsis* sp., *Trichoderma harzianum*	Soil	-	Poland	AMPA	[88]
*Pseudomonas* sp. LBr	Sludge	Glyphosate oxidoreductase and C-P lyase	Missouri	AMPA (95%), sarcosine (5%)	[94]
*Achromobacter* sp. MPS 12 A, *Ochrobactrum anthropi* GPK 3	Soil	sarcosine oxidase	Russia	Sacrosine AMPA	[96]
*Bacillus subtilis, Rhizobium leguminosarum, Streptomyces* sp.	Soil	C-P lyase and glyphosate oxidoreductase	India	AMPA and methylamine	[105]
*Fusarium oxysporum, Trichoderma viridae, Aspergillus niger*	Soil	-	Nigeria	AMPA and sarcosine	[116]
*Trichoderma viride* Strain FRP 3	Soil	-	Indonesia	-	[117]
*Aspergillus section Flavi* and *Aspergillus niger*	-	-	Argentina	-	[118]
*Ochrobactrum anthropi* S5	Soil	-	USA	AMPA	[119]
*Rhizobium meliloti* 1021	Mutation of the wild strain	sarcosine oxidase	Massachusetts	Sarcosine	[120]
*Salinicoccus* sp.	Soil	-	Iran	AMPA	[121]
*Ochrobactrum anthropi* GPK 3, *Achromobacter* sp. 16 kg	Soil	Glyphosate oxidoreductase transaminase enzyme and phosphonatase	Russia	-	[122]
*Agrobacterium radiobacter*	Wastewater	-	US	Putatively sarcosine	[123]
*Nocardia mediterranie* THS 1	-	-	India	AMPA	[124]

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
