# Peer review of "Herbicide Glyphosate: Toxicity and Microbial Degradation"

_ijerph, 2020, doi:10.3390/ijerph17207519_

Round 1

Reviewer 1 Report

The latest version of the manuscript has been significantly supplemented with data on glyphosate toxicity and its effects on organisms. These elements enrich knowledge and positively affect the level of work. For me, maybe due to the subject matter I am working on, there is still a lack of information on the actual residual levels of glyphosate in environmental samples (soil, water, plant). The presentation of data from monitoring studies will complement the subject of toxicology and may affect the correct risk assessment, the more so as articles about high exceedances of the standards (glyphosate residues) and a great risk to human health appear in the press and social media. Maybe it is worth mentioning from the point of view of science and reliable research. However, this is only a suggestion. Apart from this comment, I am not making any additional suggestions and I consider the article to be good and worth publishing

Author Response

We express our gratefulness to the learned reviewers for pointing out errors in the manuscript and for their expert & constructive suggestions. We have made our best efforts to amend the manuscript on the lines of learned reviewer’s comments. Our replies to the comments are provided below (in blue colour).

The review article “Herbicide Glyphosate: Toxicity and Microbial Degradation” discusses about glyphosate toxicity. Below are my comments:

Point 1: The latest version of the manuscript has been significantly supplemented with data on glyphosate toxicity and its effects on organisms. These elements enrich knowledge and positively affect the level of work. For me, maybe due to the subject matter I am working on, there is still a lack of information on the actual residual levels of glyphosate in environmental samples (soil, water, plant). The presentation of data from monitoring studies will complement the subject of toxicology and may affect the correct risk assessment, the more so as articles about high exceedances of the standards (glyphosate residues) and a great risk to human health appear in the press and social media. Maybe it is worth mentioning from the point of view of science and reliable research. However, this is only a suggestion. Apart from this comment, I am not making any additional suggestions and I consider the article to be good and worth publishing.

Response 1: We have compiled this review to explore the details related to glyphosate’s toxicity in all possible environment as this herbicide created global environmental challenges such as contamination of surface water, decreased soils fertility, adverse effects on soil microbiota and possible escalation in food chains. Also, we discussed the relevance of microbial biodegradation of this herbicide as crucial for maintaining the ecological equilibrium and sustainable development. We appreciate the reviewers’ remarks.

Reviewer 2 Report

The authors fulfilled all my request.

The MS is now feasible for publication.

Author Response

We express our gratefulness to the learned reviewers for pointing out errors in the manuscript and for their expert & constructive suggestions. We have made our best efforts to amend the manuscript on the lines of learned reviewer’s comments. Our replies to the comments are provided below (in blue colour).

The review article “Herbicide Glyphosate: Toxicity and Microbial Degradation” discusses about glyphosate toxicity. Below are my comments:

Point 1: The authors fulfilled all my request. The MS is now feasible for publication.

Response 1: We appreciate learned reviewer’s remarks and are thankful to him for his inputs.

Reviewer 3 Report

The review article “Herbicide Glyphosate: Toxicity and Microbial Degradation” discusses about glyphosate toxicity. Below are my comments:

There is too much redundancy in the manuscript

Page 1 line 26: replace “protecting” with a more suitable word

Page 1 line 41-43: is there any data for how much glyphosate is used globally?

Page 2 line 49: please delete “worldwide, it is most …. The sentence has already been mentioned

Page 2 line 53: in the phrase “tremendous increase” authors should include the “from” and “to”

Page 2 line 76: please specify Group 2A as probable or possible carcinogen

Page 2 lines 75-78: include year of reference

Page 3 line 118: in the sentence “it is excreted in urine …..particular environment”, authors should include a reference

Figure 2: define the terms. Example what is NADH, PSII, ETC

Page 6 line 244: replace “chief” with “main or major”

Page 7 line 308: “Biodegradation rate can be determined by using pseudo-first-order kinetics as reported in our previous studies showing  85-90% of glyphosate with half-life periods from 8.36 to 9.12 days with major metabolites”. Is this for humans or animal models?

Does glyphosate undergo one compartment or 2 compartment model?

What are the major metabolites excreted via human urine

Was glyphosate detected in urine or blood of domestic animals?

Page 8 line 17: “Glyphosate can be easily degraded by two metabolic pathways, i.e. AMPA and sarcosine via 317 glyphosate degrading bacteria”. What matrix is this?

Page 8 line 332: “AMPA is first degraded to phosphonoformaldehyde via transaminase enzyme and then further to formaldehyde via phosphonatase.” Would be easier to follow with chemical reactions or structures

Page 8 line 342: indicate the matrix in which “Glyphosate oxidoreductase (GOX) is responsible for the degradation of glyphosate to AMPA by 342 cleaving the C-N bond.” Is it mammals or insects?

Author Response

We express our gratefulness to the learned reviewers for pointing out errors in the manuscript and for their expert & constructive suggestions. We have made our best efforts to amend the manuscript on the lines of learned reviewer’s comments. Our replies to the comments are provided below (in blue colour).

The review article “Herbicide Glyphosate: Toxicity and Microbial Degradation” discusses about glyphosate toxicity. Below are my comments:

Point 1: There is too much redundancy in the manuscript

Response 1: We have compiled this review to explore the details related to glyphosate’s toxicity in all possible environment as this herbicide created global environmental challenges such as contamination of surface water, decreased soils fertility, adverse effects on soil microbiota and possible escalation in food chains. Also, we discussed the relevance of microbial biodegradation of this herbicide as crucial for maintaining the ecological equilibrium and sustainable development. To mention all the data related to this information, the review looks redundant but actually it is more extensive and informative.

Point 2: Page 1 line 26: replace “protecting” with a more suitable word

Response 2: As per the suggestion the word “protecting” is replaced with “shielding” in the mentioned sentence.

Point 3: Page 1 line 41-43: is there any data for how much glyphosate is used globally?

Response 3: The global glyphosate market was $23.97 billion in the year 2016 and is estimated to reach $34.10 billion in 2022, at a growth rate of 6.05% for the forecasted period. The details along with citation are being mentioned in the revised manuscript (Line No 52-54).

Point 4: Page 2 line 49: please delete “worldwide, it is most …. The sentence has already been mentioned

Response 4: Yes, we agree with the reviewer’s point. The sentence is modified in the revised manuscript.

Point 5: Page 2 line 53: in the phrase “tremendous increase” authors should include the “from” and “to”

Response 5: As per suggestions of the learned reviewer the sentence is modified in the revised manuscript (Line No 52-54).

Point 6: Page 2 line 76: please specify Group 2A as probable or possible carcinogen

Response 6: As per suggestions of the learned reviewer the sentence is modified as ‘The International Agency for Research on Cancer (IARC) classified herbicide glyphosate in Category ‘2a’ which specifies probable carcinogenic to humans” in the revised manuscript (Line No 75-76).

Point 7: Page 2 lines 75-78: include year of reference

Response 7: As per suggestions of the learned reviewer year of the reference is included in the revised manuscript.

Point 8: Page 3 line 118: in the sentence “it is excreted in urine …..particular environment”, authors should include a reference

Response 8: As per suggestions of the learned reviewer the reference is included in the revised manuscript.

Point 9: Figure 2: define the terms. Example what is NADH, PSII, ETC

Response 9: As per suggestions of the learned reviewer the full forms of all the abbreviations are included in the revised manuscript.

Point 10: Page 6 line 244: replace “chief” with “main or major”

Response 10: As per the suggestion the word “chief” is replaced with “major” in the mentioned sentence (line 249 of the revised manuscript).

Point 11: Page 7 line 308: “Biodegradation rate can be determined by using pseudo-first-order kinetics as reported in our previous studies showing 85-90% of glyphosate with half-life periods from 8.36 to 9.12 days with major metabolites”. Is this for humans or animal models?

Response 11: This data is related to microbial degradation by rhizobacteria isolated from the soil.

Point 12: Does glyphosate undergo one compartment or 2 compartment model?

Response 12: Various kinetic models and adsorption isotherms were found best fit to the biodegradation kinetics of glyphosates like Andrews, Han and Levenspiel, Luong, Tessier, Webb, Tseng and Wayman and Yano and Koga. Out of these pseudo order, kinetics was the best fit model for degradation studies

Point 13: What are the major metabolites excreted via human urine

Response 13: AMPA, Sarcosine, methylamine (Niemann et al. 2015)

Point 14: Was glyphosate detected in urine or blood of domestic animals?

Response 14: No it is not detected from urine or blood of domestic animals. But, it is detected from the urine of humans the data is mentioned in line No. 64-65 and 208 of the revised manuscript.

Point 15: Page 8 line 17: “Glyphosate can be easily degraded by two metabolic pathways, i.e. AMPA and sarcosine via 317 glyphosate degrading bacteria”. What matrix is this?

Response 15: Yes, Glyphosate can be easily degraded by two metabolic pathways, i.e. AMPA and sarcosine via glyphosate degrading bacteria as mentioned in line 323-324. It was noticed in the minimal media decomposition as for the degradation microbial population uses glyphosate as a sole source of energy.

Point 16: Page 8 line 332: “AMPA is first degraded to phosphonoformaldehyde via transaminase enzyme and then further to formaldehyde via phosphonatase.” Would be easier to follow with chemical reactions or structures

Response 16: Figure 3 depicts the whole degradation pathway along with enzymes.

Point 17: Page 8 line 342: indicate the matrix in which “Glyphosate oxidoreductase (GOX) is responsible for the degradation of glyphosate to AMPA by 342 cleaving the C-N bond.” Is it mammals or insects?

Response 17: It’s a bacterial metabolic process. Glyphosate oxidoreductase is responsible for the degradation of glyphosate to AMPA by cleaving the C-N bond by bacterial species under minimal media. Which makes glyphosate to used as a carbon and nitrogen source by the respective bacteria.

Round 2

Reviewer 3 Report

The review article “Herbicide Glyphosate: Toxicity and Microbial Degradation” discusses about glyphosate toxicity. Below are my comments:

Page 2 line 75: authors mentioned information from IARC and yet provided the reference by Singh, S.; Kumar, V.; Datta, S.; Wani, A.B.; Dhanjal, D.S.; Romero, R.; Singh, J. Glyphosate uptake, 431 translocation, resistance emergence in crops, analytical monitoring, toxicity and degradation: a review. 432 Environ. Chem. Lett. 2020, 18, 663–702.

Usually, authors provide a cover letter discussing the point by point comments raised by reviewers or some form of justification. I couldn’t find responses to some of the comments or questions I previously raised. For instance:

  1. Does glyphosate undergo one compartment or 2 compartment model?
  2. What are the major metabolites excreted via human urine
  3. Page 8 line 332: “AMPA is first degraded to phosphonoformaldehyde via transaminase enzyme and then further to formaldehyde via phosphonatase.” Would be easier to follow with chemical reactions or structures

Author Response

We express our gratefulness to the learned reviewers for pointing out errors in the manuscript and for their expert & constructive suggestions. We have made our best efforts to amend the manuscript on the lines of learned reviewer’s comments. Our replies to the comments are provided below.

Point 1: Page 2 line 75: authors mentioned information from IARC and yet provided the reference by Singh, S.; Kumar, V.; Datta, S.; Wani, A.B.; Dhanjal, D.S.; Romero, R.; Singh, J. Glyphosate uptake, translocation, resistance emergence in crops, analytical monitoring, toxicity and degradation: a review. Environ. Chem. Lett. 2020, 18, 663–702.

Response 1: As per suggestions of the learned reviewer the reference for IARC as "International Agency for Research on Cancer (2017). Some organophosphate insecticides and herbicides. IARC monographs on the evaluation of carcinogenic risks to humans, 112, 1-452” is included in the revised manuscript.

Point 2: Usually, authors provide a cover letter discussing the point by point comments raised by reviewers or some form of justification. I couldn’t find responses to some of the comments or questions I previously raised.

Response 2: We wish to inform you that we have attached the response to reviewer’s comment along with Author's Notes to Reviewer in the provided column of the revision submission portal, but was not available to the reviewer.

Point 3: Does glyphosate undergo one compartment or 2 compartment model?

Response 3: Glyphosate undergo one-compartment model with an elimination rate constant of Kel = 0.021 hour-1 when 100 mg/kg bw) of glyphosate concentrations in plasma in Sprague-Dawley rats (Bernal et al., 2010)

Point 4: What are the major metabolites excreted via human urine?

Response 4: AMPA and glyphosate itself are the major metabolites detected in human urine samples (Niemann et al 2014; Connolly et al. 2018; Soukup et al. 2020)

Point 5: Page 8 line 332: “AMPA is first degraded to phosphonoformaldehyde via transaminase enzyme and then further to formaldehyde via phosphonatase.” Would be easier to follow with chemical reactions or structures

Response 5: As per suggestions of the learned reviewer an illustration is included as Figure 3 in the revised manuscript.

This manuscript is a resubmission of an earlier submission. The following is a list of the peer review reports and author responses from that submission.

Round 1

Reviewer 1 Report

Glyphosate - behavior in the environment, and especially impact on human health is currently a very popular and current topic of many scientific and popular articles. The fate of this substance is uncertain and many groups are lobbying for its removal from the market of many countries (including the European Union). Glyphosate is a topic for a book, not just an article. The article has a limited number of pages, however, writing in the title "risk assessment and fate in the environment" authors should significantly extend the scope of the article. In my opinion, some important information and data are missing.

Introduction:

Expand information on the use of glyphosate - it's not just GMO crops. In Europe, where GMO crops are significantly restricted or banned, glyphosate is used as a desiccant, which is the greatest threat. Due to the short period between desiccation and crop harvesting, we may have a problem with too much residues of this substance (above the permissible level).

Expand information on the impact of glyphosate on human health

Lack of information on legislative problems (e.g. in the European Union - work on withdrawing this substance from the market)

  1. Degradation Kinetics

New chapter title (suggestion): Degradation Kinetics and glyphosate residues

Describe (rather in a general way) methods of analysis of glyphosate residues (analytical apparatus, level of determination, occurring problems) - such information was recorded in the summary, and it is not in the text of the article

Add examples of residue test results (soil, water, plant) - results, exceeding standards (MRL)

Add the test examples to the kinetics together with the t1/2 value

Please explain – lines 156-157 (“… continued use of this very herbicide….

The literature cited is current and well-chosen

Final assessment: Reconsider after major revision

Author Response

Sub: Replies to the reviewers' comments on manuscript # ijerph-871641

Point 1: Glyphosate - behavior in the environment, and especially impact on human health is currently a very popular and current topic of many scientific and popular articles. The fate of this substance is uncertain and many groups are lobbying for its removal from the market of many countries (including the European Union). Glyphosate is a topic for a book, not just an article. The article has a limited number of pages, however, writing in the title "risk assessment and fate in the environment" authors should significantly extend the scope of the article. In my opinion, some important information and data are missing.

Response 1:

As suggested by the learned reviewer, the whole manuscript is revised specifying the important information and data related to risk assessment of glyphosate in the environment. All the corrections have been highlighted yellow in the revised manuscript.

Point 2:

Introduction:

Expand information on the use of glyphosate - it's not just GMO crops. In Europe, where GMO crops are significantly restricted or banned, glyphosate is used as a desiccant, which is the greatest threat. Due to the short period between desiccation and crop harvesting, we may have a problem with too much residues of this substance (above the permissible level).

Response 2:

As suggested by the reviewer, the information related to use of glyphosate is updated in the revised manuscript. Also, to clarify this point we need to add that glyphosate resistant cropping systems are now used worldwide, which includes Zea mays (maize) Gossypium hirsutum (cotton) Glycine max (soybean), and Brassica napus (canola) also. These crops are having no direct impact or risk when transgenes of glyphosate resistant are introduced into wild type populations.

Point 3:

Expand information on the impact of glyphosate on human health

Response 3:

As suggested by the learned reviewer, extended information related to the impact of glyphosate on human health has been now incorporated in the revised manuscript (Line 201-242).

Point 4:

Lack of information on legislative problems (e.g. in the European Union - work on withdrawing this substance from the market)

Response 4:

As per suggestion of the reviewer, this information has been incorporated in the introduction section of the revised manuscript highlighted yellow (Line 72-77).

Point 5:

4. Degradation Kinetics

New chapter title (suggestion): Degradation Kinetics and glyphosate residues

Response 5:

As suggested by the learned reviewer, section 4. Degradation kinetics is updated to Degradation kinetics and glyphosate residues in the revised manuscript (Line 297-314).

Point 6:

Describe (rather in a general way) methods of analysis of glyphosate residues (analytical apparatus, level of determination, occurring problems) - such information was recorded in the summary, and it is not in the text of the article

Response 6:

As suggested by the reviewer, this has been mentioned in section 2.3 and highlighted yellow in the revised manuscript.

Point 7:

Add examples of residue test results (soil, water, plant) - results, exceeding standards (MRL)

Response 7:

As suggested by the reviewer in section 2.3, examples of residue test results (soil, water, plant) - results, exceeding standards (MRL) are added in the revised manuscript (Line 201-242).

Point 8:

Add the test examples to the kinetics together with the t1/2 value

Response 8:

As suggested by the reviewer the test examples to the kinetics together with the t1/2 value are added in the revised manuscript (Line 310-312).

Point 9:

Please explain – lines 156-157 (“… continued use of this very herbicide….

Response 9:

The word ‘continuous’ has been replaced by ‘incessant’ for better clarity. This statement says that if glyphosate application remains in continuity, it leads to reduction in nodule formation and nitrogen fixation process (Line 150).

Point 10:

The literature cited is current and well-chosen

Response 10:

We appreciate reviewers’ remarks

Point 11:

Final assessment: Reconsider after major revision

Response 11:

As suggested by the learned reviewer, the whole manuscript is revised according to the valuable suggestions. All the corrections have been highlighted yellow in the revised manuscript.

Reviewer 2 Report

This paper on glyphosate tries to cover a lot of ground. Firstly the title is strange and misleading since there is no real section on risk assessment. There is a rather vague section on different toxicity mechanisms – but very little on exposure and without that risk assessment would be not meaningful. There are many issues with the paper – it tries to cover too much ground and is very superficial adding very little new knowledge.

Probably the stronger part of the paper is on the degradation pathways and maybe if the paper is to be revised this is what the title of this article.

Specific comments (examples)

There are many issues with language/English. Even consistency (if so many authors contribute one could expect a bit of checking these things.   Glyphosate is spelled with capital or not …  

There has been mention of how much glyphosate ends up in the environment, as well as its detection in surface water and even urine, but no actual values reported/properly reviewed or anything about spatio/temporal trend and whether exposure has been increasing or not.

In the toxicity section (86) there is mention of glyphosate's toxicity to organisms, but no reference to levels (LC50, EC50 etc), how much glyphosate is required to cause toxic effects.  This is thus meaningless.   (Also no discussion of toxicity of surfactants that are used and how they may affect some of the published information.)

Sentences in lines 104-106 need references

110 needs references, and says overuse of glyphosate causes toxicity, but how much (in numbers) is overuse?   116 needs references

There is mention of health risks associated with glyphosate (70, 117, 304) but not much further than that. What about exposure assessment.

Author Response

Sub: Replies to the reviewers' comments on manuscript # ijerph-871641

Point 1:

This paper on glyphosate tries to cover a lot of ground. Firstly, the title is strange and misleading since there is no real section on risk assessment. There is a rather vague section on different toxicity mechanisms – but very little on exposure and without that risk assessment would be not meaningful. There are many issues with the paper – it tries to cover too much ground and is very superficial adding very little new knowledge.

Response 1:

As suggested by the learned reviewer, the whole manuscript is revised specifying the important information and data related to risk assessment of glyphosate in the environment and included under sections 2.1. Toxicity against non-target plants, 2.2. Toxicity against Animals and 2.3. Toxicity on human health in the revised manuscript.

Point 2:

Probably the stronger part of the paper is on the degradation pathways and maybe if the paper is to be revised this is what the title of this article.

Response 2:

As suggested by the reviewer, we updated the manuscript on the lines of comments and elaborated the risk assessment section. If learned reviewer still thinks the title require to change, we will do it accordingly.

Point 3:

Specific comments (examples)

There are many issues with language/English. Even consistency (if so many authors contribute one could expect a bit of checking these things.   Glyphosate is spelled with capital or not …

Response 3:

As per suggestions, the language of the whole MS is corrected to make readers better understand the contents and the consistency in writing is also checked. All the changes made are yellow high lightened in the revised manuscript.

Point 4:

There has been mention of how much glyphosate ends up in the environment, as well as its detection in surface water and even urine, but no actual values reported/properly reviewed or anything about spatial/temporal trend and whether exposure has been increasing or not.

Response 4:

As suggested by the reviewer, the manuscript is revised specifying quantity of glyphosate ends up in the environment, as well as its detection in surface water and even urine, but no actual values reported/properly reviewed or anything about spatial/temporal trend and whether exposure has been increasing or not under section 2.3 for the revised manuscript (Line 201-242)

Point 5:

In the toxicity section (86) there is mention of glyphosate's toxicity to organisms, but no reference to levels (LC50, EC50 etc), how much glyphosate is required to cause toxic effects.  This is thus meaningless.   (Also no discussion of toxicity of surfactants that are used and how they may affect some of the published information.)

Response 5:

As suggested by the reviewer, some toxicity studies depicting LD50 have been added (Line 201-242)

Point 6:

Sentences in lines 104-106 need references

Response 6:

As suggested by the reviewer, reference has been included in the revised manuscript.

Point 7:

110 needs references, and says overuse of glyphosate causes toxicity, but how much (in numbers) is overuse?

Response 7:

It has been mentioned for non-target organisms. Because there are a variety of non-target organisms for glyphosate, a specific value in number cannot be given. However, a reference for the same as suggested by the reviewer is incorporated.

Point 8:

116 needs references

Response 8:

As suggested by the author, the references have been added.

Point 9:

There is mention of health risks associated with glyphosate (70, 117, 304) but not much further than that. What about exposure assessment.

Response 9:

As per suggestions of the reviewer, the health risk associated data has been incorporated (Line 201-242)

Reviewer 3 Report

The MS is an interesting review focusing on the environmental fate of a trending pesticide: Glyphosate.

It is well organized, but I suggest the authors to check the text for missing words along it and to have the MS revised by an mother language English revisor.

Also, I would like the authors to state in extensu the meaning of the acromyn EPSPS at line 52, where it is firstly cited.

This said, if English will be improved I consider the MS as suitable for publication.

Author Response

Sub: Replies to the reviewers' comments on manuscript # ijerph-871641

Point 1:

The MS is an interesting review focusing on the environmental fate of a trending pesticide: Glyphosate.

Response 1:

We appreciate reviewers’ remarks

Point 2:

It is well organized, but I suggest the authors to check the text for missing words along it and to have the MS revised by an mother language English revisor.

Response 2:

As per suggestions, the language of the whole MS is corrected to make readers better understand the contents and the consistency in writing is also checked. All the changes made are high lightened yellow in the revised manuscript.

Point 3:

Also, I would like the authors to state in extensu the meaning of the acromyn EPSPS at line 52, where it is firstly cited.

Response 3:

As per suggestions of the reviewer, the acronym of ‘EPSPS’ is mentioned at the firstly cited place.

Point 4:

This said, if English will be improved I consider the MS as suitable for publication.

Response 4:

As per suggestions, the language of the whole MS is corrected to make readers better understand the contents. The changes made are highlighted yellow in the revised manuscript.

Round 2

Reviewer 1 Report

The revised manuscript includes all the corrections suggested by the reviewer. The current version is "more complete" and contains most of the information that should be included in a review article. A good review of the literature and a good, concise form should guarantee a large number of citations of the publication

I have no additional comments - the article is suitable for publication in a journal